# Pragmatic Systemic Solutions to the Wicked and Persistent Problem of the Unprofessional Disruptive Physician in the Health System

**DOI:** 10.3390/healthcare11172455

**Published:** 2023-09-01

**Authors:** Carmelle Peisah, Betsy Williams, Peter Hockey, Peter Lees, Danette Wright, Alan Rosenstein

**Affiliations:** 1Faculty of Medicine and Health, The University of Sydney, Sydney, NSW 2006, Australia; peter.hockey@health.nsw.gov.au; 2Discipline of Psychiatry and Mental Health, Faculty of Medicine, University of New South Wales, Sydney, NSW 2052, Australia; 3Professional Renewal Center^®^, Lawrence, KS 66049, USA; bwilliams@prckansas.org; 4Continuing Medical Education Wales Behavioral Assessment, Lawrence, KS 66049, USA; 5Department of Psychiatry, School of Medicine, University of Kansas, Kansas City, KS 66045, USA; 6Western Sydney Local Health District, Sydney, NSW 2145, Australia; danette.wright@health.nsw.gov.au; 7Faculty of Medical Leadership and Management, London WC1R 4SG, UK; peter.lees1@fmlm.ac.uk; 8School of Medicine, Western Sydney University, Sydney, NSW 2560, Australia; 9Internal Medicine, Health Care Behavioral Management, San Francisco, CA 94118, USA; ahrosensteinmd@aol.com

**Keywords:** disruptive, physicians, healthcare, systems, professionalism, physicians’ health

## Abstract

We have always had and will always have “disruptive” or “dysfunctional” doctors behaving unprofessionally within healthcare institutions. Disruptive physician behaviour (also called “unprofessional behaviour”) was described almost 150 years ago, but remains a persistent, wicked problem in healthcare, largely fuelled by systemic inaction. In this Commentary, we aim to explore the following aspects from a systemic lens: (i) the gaps in understanding systemic resistance and difficulty in addressing this issue; and (ii) pragmatic approaches to its management in the healthcare system. In doing so, we hope to shift the systemic effect from nihilism and despair, to one of hopeful realism about disruptive or unprofessional behaviour. We suggest that solutions lie in cultural change to ensure systemic awareness, responsiveness and early intervention, and an understanding of what systemic failure looks like in this context. Staff education, policies and procedures that outline a consistent reporting and review process including triaging the problem, its source, its effects, and the attempted solutions, are also crucial. Finally, assessment and intervention from appropriately mental-health-trained personnel are required, recognising that this is a complex mental health problem. We are not doing anyone any favours by ignoring, acting as bystanders, or otherwise turning a blind eye to disruptive or unprofessional behaviour; otherwise, we share culpability.

## 1. Introduction

We have always had and will always have “disruptive” or “dysfunctional” doctors behaving unprofessionally within healthcare institution. Disruptive physician behaviour (DPB) was described almost 150 years ago, while the term disruptive physician was coined in 1995 [1,2]. The problem of DPB is so ubiquitous [3,4,5,6] and manifests itself in such myriads of ways that 207 unique terms have been used to describe it [2]. Behaviours associated with the term include, but are not limited to, those openly aggressive, such as predatorial sexual behaviours or verbal or physical abuse of colleagues or throwing instruments or equipment; and those more “passive” such as non-compliance with protocols, hiding errors, threatening colleagues, and committing slander and reputational damage, betraying confidentiality, making vexatious complaints, or weaponizing emails or quality and safety and accreditation processes [3,4,5,6].

Regardless of the elusiveness of the concept, the core aspect critical to its recognition and management, is that it causes disruption to the healthcare system. DPB is counterproductive and contributes to poorer (i) teamwork, (ii) morale, (iii) job satisfaction, (iv) communication, (v) work–life balance and (vi) patient outcomes [1,2,3,4,5,6,7]. In short, disruptive and unprofessional behaviour fosters a culture of incivility and disrespect, all of which pose a threat to both patient safety and staff wellbeing [7].

DPB is a systemic problem that requires a systemic approach. A systemic approach, based on the Systems Theory, implicitly includes the individual, but offers a broader framework for seeing DPB within the complex and recursive inter-relationships within modern health systems. A systemic approach implicitly offers a more complex mapping of the relationships between the multiple components of the health system, including the multiple systemic factors that contribute to DPB. The kind of systemic factors at play here include (i) the rules of the health system (e.g., “We have never heard of DPB” or “We tolerate DPB” or “We can’t touch her, she is too powerful to take on”); (ii) the history of the health system; and (iii) the functioning and structure of the health system, and relationships within, including alliances and conflicts. This broader perspective per se might help generate more solutions than reductionistic approaches to the causes of DPB, such as the simplistic linear, “individual x causes y” approach, which may limit options in response and quelch hope. It is from this perspective that we write this Commentary. Moreover, while we recognise that dysfunctional behaviour in health systems is not limited to physicians, over 80% of notifications for dysfunctional behaviour relate to physician behaviour, while 50% of notifications relate to nurses, with minimal study of other health disciplines [8]. We thus focus on the specific problem here as it relates to physician behaviour.

Despite concerted efforts of the profession, the problem of DPB persists and will not go away [3,4,5,6], nor do we predict that it will. There will always be difficult doctors, lawyers and politicians and other professionals in high-stake, high-profile professions whose behaviour contributes to havoc in their workplace. It is unrealistic to approach this very human problem with the hope of eradication, but more realistic to do so with the goal of identifying the underlying contributory factors determining what is fixable and otherwise considering amelioration and damage control.

From a systemic lens, we aim to explore: (i) some of the gaps in understanding and how these might contribute to resistance and difficulty in addressing this issue; and (ii) pragmatic approaches to its management in the health system. We do so from a multi-disciplinary perspective informed by the frontline, leadership and academic expertise of a (i) psychiatrist with family and systems training, running a medical Professional and Systems Support Unit in an Australian public hospital setting (CP) [9,10]; (ii) a clinical psychologist and researcher with training in neuropsychology and public health who provides assessment, treatment, and remedial continuing professional support to healthcare professionals referred for disruptive and unprofessional behaviour in the United States (US) (BW) [11]; (iii) a physician leader with national experience as a Postgraduate Dean and Responsible Officer in the United Kingdom (UK) and executive-level experience in the Australian healthcare system within Medical Administration, Education, Innovation, and Quality and Safety (PH); (iv) a neurosurgeon and experienced Medical Director in the United Kingdom (UK) National Health Service (NHS) and former Chief Executive, UK Faculty of Medical Leadership and Management (PL); (v) a general and colorectal surgeon and head of surgery with senior leadership role in an Australian public teaching hospital (DW); and (vi) an internist and senior medical management consultant with extensive experience in case management, data-based clinical decision support, disruptive staff relationships, and physician engagement strategies in the US (AR) [3,4,5,6]. As such, we write this from the perspective of three different countries grappling with the problem for decades. Our aims in writing this Commentary are to collectively increase understanding of DPB, its systemic drivers and systemic solutions, and in doing so, shift the systemic affect from nihilism and despair regarding DPB, to one of hopeful realism.

## 2. Better Understanding of Disruptive Physicians: Individual Physician Causes

Disruptive physician behaviour has sometimes been defined as a “a practice pattern of personality traits” [12]. Although maladaptive personality is often assumed to be a common cause of DPB, it is not the only cause of DPB. Neff, reporting data on 202 physicians referred to the Professional Assessment Program, Abbott Northwestern Hospital US, with disruptive behaviour, found that 27% had a personality disorder or traits, while 78% had a primary psychiatric disorder (DSM Axis I). Of those with a psychiatric diagnosis, 40% suffered from major depression, 27% from alcoholism/chemical dependency, 6% from a sexual disorder, 6% from a bipolar illness, and 2% were diagnosed with obsessive–compulsive disorder [13].

Similarly, in a chart review of physicians referred for disruptive behaviour to the Florida Professionals Resource Network, a US physician health program that monitors healthcare professionals, Merlo et al. found that 55% had one psychiatric diagnosis while 20% had multiple diagnoses, approximately one third had a diagnosis, traits, or features from one personality disorder cluster, while 39% had multiple-personality diagnostic clusters. Fifty percent had both psychiatric and personality diagnoses, traits, and/or features noted. In terms of personality, the majority were diagnosed as having either Cluster B (48%) or Cluster C traits (50%), while 2% were assigned a diagnosis of a personality disorder not otherwise specified. Co-morbid diagnoses included mood disorders (24%), anxiety disorders (11%), adjustment disorders (30%), impulse control disorders (30%), substance use disorders (6%), and sexual disorders (2%) [14].

Notably, while we seek to illustrate the diversity of mental health disorder beyond personality disorder, we emphasise that personality dysfunction in doctors manifesting as DPB cannot be ignored, nor can the underlying role of trauma-related disorders and adverse childhood experiences [9,15,16] in driving such behaviour. For example, in the aforementioned study by Neff, 66% of the physicians reported emotional neglect or abuse, and 21% physical abuse including sexual abuse [13]. Importantly, notwithstanding the complexity illustrated by these data, we emphasise that if recognised, many of these conditions are ameliorable or treatable.

We acknowledge that the data from these studies are skewed by their context, namely doctors referred to monitoring or regulatory agencies. Other causes of DPB unaddressed in these cohorts are neurodegenerative and neurodiversity disorders. Neurodegenerative disorders such as mild cognitive impairment or dementia (or major and minor neurocognitive disorder), alone or comorbid with other disorders, are also important to consider as potential causes of DPB presenting for the first time in a doctor over 50 [17].

Little studied in doctors is neurodiversity, and in particular, autistic spectrum disorder (ASD). It is not uncommon for ASD to be identified later in life, especially as those affected often have high intellectual functioning with strong cognitive and verbal abilities. These individuals often have traits vital for skilled medical practitioners such as responsiveness to structure, attention to detail and unique creative skills, but also frequently experience relationship difficulties manifested as poor or unprofessional behaviours. Consideration of ASD with subsequent assessment and diagnosis can help both the individual and the system in which they are working to make appropriate modifications to working patterns and support [18,19].

Beyond frank mental health disorders, DPB can also be driven by more minor symptomatic manifestation of what Rosenstein describes as “life in the fast lane” of medicine, namely stress, frustration, dissatisfaction, and burnout from overburdened under-resourced health systems [20,21,22]. Notably, the most common contexts for DPB are complex health environments with high care levels and associated emotional and physical burden, namely intensive care, surgical and emergency departments [8]. These manifestations of system disintegration are particularly important to consider given the developing literature around increased mental health concerns and burnout in physicians since the COVID-19 pandemic. Recent survey results show that health systems across the world are facing a crisis of physically and mentally overburdened physicians, culminating in massive attrition of the workforce, further exacerbating resource issues and burden [23,24]. Keeping true to our circular, systemic perspective, it is important to consider both the failure of the medical workforce to respond to such threats [23,24] and the effect of such systemic breakdown on physicians, mediated by their own individual resilience and vulnerability, or their “capability” [25] in facing such chaos. Such capability factors can include a lack of knowledge around elements of the changing landscape of medicine such as more current expectations around behavioural comportment, the American Board of Medical Specialties or CanMeds competencies, as well as skill deficits related to personality vulnerabilities as previously discussed, manifesting in poor emotional competency and coping skills [26].

Another important consideration and sometime neglected contributory factor to DPB is medical illness [11,14], certainly a contributor to impairment in the older physician [17]. While studies of the incidence of physical illness among practicing physicians are lacking, a reasonable estimate is that at least 10% of physicians must restrict their practice for several months or more during their career because of a disabling physical illness such as diabetes, heart disease, or surgical procedures [27]. Data on a sample of 117 physicians referred for concerns related to unprofessional behaviour found that referred physicians had significantly poorer health than a comparison sample of physicians with the five most frequent medical conditions being previously undiagnosed or poorly treated hypertension and sleep apnoea, obesity, hyperlipidaemia, and pain [28].

We make this complicated because it is, and because ironically, the more contributors to the problem, the more there are potential targets for remediation, again justifying our position of hope.

## 3. Better Understanding of Disruptive Physicians: Effects of DPB

Just as unhelpful as a reductionistic approach to individual physician causes of DPB, is the often reductionistic approach to its systemic effects. Traditionally, DPB has been defined by its effect on patient care and safety: The disruptive actions can result in delay of appropriate treatment, injury, or death of patients [29] (p. 194). It is rightly so that we emphasise this. However, this narrow focus fails to recognise the equally damaging effects of DPB on colleagues and team performance. In addition to fostering medical errors, patient dissatisfaction, preventable adverse outcomes, and increased costs of care, DPB threatens the performance of the health care team. DPB may also contribute to attrition of valuable staff, causing experienced and valued clinicians, managers and administrators and managers to seek new positions in more professional environments [30,31,32]. Our own observation and experience suggest that DPB culminates in harm to colleagues, often the very people seeking to enforce zero tolerance and draw boundaries around unacceptable behaviour [10].

Clearly, patient care, teamwork and safe workplaces are all equally important, being inextricably linked, with safety and quality being dependent on communication, a collaborative work environment and the ability to put constraints on unacceptable behaviour [33,34]. Although a seemingly obvious point, the pragmatic import of this is that the system must act, rather than sit on its hands, when it receives information about DPB impacting staff, just as it does when it receives complaints from patients and relatives. DPB that damages colleagues is equally as important as DPB that damages patients. It is still commonplace that technical excellence or lack of impact on patient safety are offered as rationales for ignoring poor behaviour with no acknowledgement (and perhaps ignorance) of the impact of DPB on co-workers.

## 4. Systemic Responses and Attempted Solutions to Date

The first and perhaps the most important systemic response to DPB since its “discovery,” was naming it and calling it out, as did the Federation of State Medical Boards (FSMB) in 2000 by acknowledging the importance of addressing DPB [35]. Recently, this has become more nuanced, with the term “unprofessional behaviour” suggested by the Joint Commission, an independent, not-for-profit organization agency in the United States that sets standards and accredits healthcare entities [32]. Again, we note that the issue of naming DPB or unprofessional behaviour might seem trivial, but there are still in 2023, health systems globally who have never heard of either term.

The second, and equally important, systemic response was articulating rules and boundaries around DPB, while, at the same time, describing its corollary, acceptable behaviour [12]. In 2008, the Joint Commission included disruptive and inappropriate behaviours within its Leadership Standards and clearly established behavioural rules and boundaries for medical systems [30,31]. At the same time as proscribing DPB, systems need to prescribe appropriate behaviour, including management and leadership behaviours expected of good doctors, articulated by the Faculty of Medical Leadership and Management in the UK [36]. Without such, systems are ruleless with regards to DPB, and lack both boundaries and benchmarks for measurement and improvement. How can those who lack internal loci of control or appropriate social and behavioural skills (for whatever reason), behave appropriately if nobody has articulated what behaviour is expected of them, or the consequences of such behaviour? If we are not explicit about the behaviours we expect, is it not surprising that some do not conform to undefined norms.

The third systemic response has been deciding whose job it is to implement these codes. For example, in 2008, the aforementioned Joint Commission tasked healthcare organizations and hospital leadership with enforcement and the development of Codes of Conduct defining acceptable and disruptive and inappropriate behaviours and the implementation of processes for managing these behaviours [30,31]. Such codes include recommendations for a culture of zero tolerance for DPB, defining appropriate behavioural standards, and holding individuals accountable for their behaviours [6,12].

Finally, many systems have progressed beyond mere policy, with a range of systemic responses to DPB in place for years. The whole gamut of skill impairment is at play here including lack of emotional intelligence, insight/awareness, boundaries, conflict resolution, social skills, and communication. Accordingly, a range of remedial programs, education and training exist including training in communication, team collaboration, diversity management, cultural competency, emotional intelligence, and conflict management [6,11,25,37].

## 5. Barriers

Notwithstanding these efforts, the DPB problem persists [6]. In answering why this is so, Rosenstein has identified a range of barriers to tackling this problem, many of which are systemic failures with organisational responsiveness being key to the solution [6]. Organizational failure may manifest itself in a code of silence or reluctance to act. Behind such reluctance is often a fear of legal retribution from the physician with DPB, or because the physician is valuable to the organization, or to those responsible for action, whose judgment may be contaminated by bias, or conflict of interest [6]. Associated with this are gaps in policy and process failure. Systemic inaction, delayed response or even responses that fuel DPB can be at several levels, including at the organization itself, at the supra-organisational level or from satellite systems, such as Specialist Training Colleges. Responses that fuel DPB include taking on vexatious feedback and scapegoating of leaders at face value, without confronting the basis for such.

Problematic behaviour that is not appropriately addressed causes a myriad of problems. It can also make extinguishing the problematic behaviour more challenging. There are countless examples of systems that inadvertently reinforce disruptive behaviour, by inadvertently providing the identified physician the desired self-serving outcome [34]. An important element to shifting performance is feedback, both corrective feedback and feedback that reinforces a job well done. Systems often have a lack of awareness of how a lack of response contributes to the ongoing promotion of a culture of incivility and disrespect [7]. Inaction or late action may also allow the disruptive physician to form dysfunctional alliances within the system, joining with others whose purposes are equally served in doing so, allowing the DPB to gain momentum or become encrusted in the system (see Appendix A Figure A1 Flow chart).

## 6. Potential New Solutions

We emphasise that the solution to a system disruptor cannot solely lie with the physician displaying the DPB, nor with the targets of DPB, in some cases, the “victims” of such. Most importantly, the voice of one person, regardless of their seniority, is unlikely to be heard. Moreover, the solo whistle-blower is at risk of vindictive retaliation and litigation.

Cultural change is required to ensure systemic awareness and responsiveness [6,11,34]. We build on existing recommendations [6], to effect cultural changes. First and foremost, as stated earlier, we must raise systemic awareness from the top of the healthcare organisation down. The organisation/hospital executive and administration, and human resource department must all be educated about DPB. Awareness raising of the risks of inaction towards DPB [6,11,34], with its converse, prevention, and early intervention [38] may galvanise earlier action from the hospital administration. Risks of inaction include at minimum, compromised patient satisfaction and organizational reputation, with resultant litigation; and at maximum, risks to patient safety with serious adverse events including mortality [6,29,30,31,32,33,39,40]. Other systemic risks include effects on organizational morale, recruitment and retention, and care efficiency with compromised process flow and productivity [6,29,30,31,32,33,39,40], all of which carry additional financial risks [41]. One study of a 400-bed hospital estimated the combined costs of DPB attributable to medication and procedural errors, and staff turnover, were in excess of 1 million US dollars [42]. Besides these systemic risks, an oft-neglected risk conferred by failure to address DPB is the unaddressed and potential deterioration of mental health and wellbeing of the doctor with DPB, themselves often otherwise valuable assets of the system.

We would add that awareness and understanding of DPB must be echoed across all systems, including not only the system in which the DPB is occurring, but in supra- systems and satellite systems (see Appendix A Figure A1 Flow chart). Absolutely essential to all systems must be protection against intimidation or retribution (including with reverse vexatious complaints) of those who report or cooperate in any DPB investigation. Such protection may include, but should not be limited to, non-retaliation clauses and general support [43].

Secondly, staff education and training, and the soliciting of champions from within staff are required to improve engagement, prevention, recognition, documentation, and accountability. We note here that intrinsic to these solutions is medical education, which has only very recently ventured into behaviours and leadership, with much of its focus on technical excellence in all its forms. This may go a long way to explain why poor behaviour has been tolerated for so long, it simply did not feature in the mandate or curricula of educational bodies. To that end, Hickson et al. have described the Vanderbilt University School of Medicine (VUSM) curriculum: “Every physician needs skills for conducting informal interventions with peers…. Physician leaders receive skills training for conducting higher-level interventions. No single strategy fits every situation, so we teach a balance beam approach to understanding and weighing the pros and cons of alternative intervention-related communications. Understanding common excuses, rationalizations, denials, and barriers to change prepares physicians to appropriately, consistently, and professionally address the real issues” [38] (p. 1040). Not only do we have a clearly articulated model for education around responding to DPB with this VUSM program [38]; there is evidence that professional behaviour can be measured, and remedial strategies taught through continuing medical education programs [44]. Most importantly, systems need to understand who needs to be educated about what, and in doing so, who needs to be empowered. Pathologising whistle-blowers by sending them to communication training or mediation or asking them to account for themselves in response to retaliatory accusations of bullying (as we have witnessed) is not appropriate. Triaging the problem, its source, its effects, and its attempted solutions are therefore crucial (see Appendix A Figure A1 Flow Chart).

Thirdly, policies and procedures that outline a consistent reporting and review process are required, but these must be supported by audits of adherence. The largest system intervention of policies and procedures in medical regulation for 150 years in the UK—Revalidation—was formally introduced by the General ‘Medical Council in 2012. The intent—to ensure doctors are “up to date” and “fit to practise”—is an unarguable aspiration rightly expected by all patients and their caregivers. The operationalisation of the Revalidation system is carried out through a formalised system of annual appraisal undertaken by trained appraisers directly accountable to their organisation’s most senior medical leader (known as the Responsible Officer). The process of appraisal allows doctors to demonstrate that they believe in the values and principles of their profession, follow contemporary guidance and importantly that they reflect on their practice and achievements. While the Revalidation system has at its core a positive intent to support standards and improvement, it has also provided a lever for Responsible Officers to manage and hold to account those doctors whose behaviours and professionalism fall short of accepted standards. The fact that the Responsible Officer is a local and respected senior medical leader with statutory responsibility for ensuring professional standards and direct accountability to the medical regulator brings a visible and local human face to regulation. This role and attendant appraisal system brings a clear legitimacy to being able to name and tackle dysfunctional behaviour. This is a good example of how a systematic and mandatory process can provide a health system an ability to hold doctors to account for identified poor behaviours. We note that in the US there has been a call to expand the number and capacity of assessment programs that deal with physicians with behavioural problems [27]. Perhaps in other health settings we could follow with establishing such bespoke programs in the first place.

Fourthly, assessment of the potential roles of physical and mental health issues must always be considered with interventions from the appropriate medical and/or mental health professional readily available. Disruptive physician behaviour is a complex issue with multiple contributory factors. To support the best outcome there must be access to a gamut of resources including physician skill training, behavioural interventions, medical or psychiatric consultation, and mental health treatments that can be tailored to each individual DPB context.

Finally, there must be some reflection about the way the system attracts, breeds, and fuels this behaviour. It could be argued that first and foremost, the medical profession attracts certain personality styles prone to such behaviours, the so called “dark triad” traits of narcissism, Machiavellianism, and psychopathy, albeit in a minority [45]. Such personality styles render physicians prone to aggression and hostility under conditions of ego threat (i.e., uncertainty, any questioning or threat to self-esteem) [46,47]. Further, medical training is perfectly designed to develop a range of skills that on the one hand, foster a drive for excellence, to advance personal knowledge, continuous improvement, efficiency, and perfection, namely ‘expert’ action logic [48,49]. The unchecked downside of the “expert” is an undue certainty that they are right; contempt for those they see as less competent; a tendency to micromanage and failure to see the bigger picture. These, coupled with little understanding or desire for emotional intelligence, can make so-called “experts” poor managers or leaders. Recognizing both individual and system factors as Leape and colleagues aptly note, “disrespectful behaviour is rooted, in part, in characteristics of the individual, such as insecurity or aggressiveness, but it is also learned, tolerated, and reinforced in the hierarchical hospital culture” [7].

We need to understand what systemic failure looks like in the context of DPB. Red flags for systemic failure include late identification of DPB, failure to respond to longstanding DPB, undue empowerment of the physician with DPB, unchecked scapegoating or victimisation of the targets of DPB and the development of alliances involving groups of physicians with DPB. Systemic red flags mandate that we go back to the drawing board and assess where the gaps are with policy, standards, and action across all systems (see Appendix A Figure A1 Flow chart).

A recent focused review in the context of 20 years of experience of assessing, treating, and remediating unprofessional behaviour highlighted that disruptive/unprofessional behaviour needs to be considered in the context of the entire system from the individual and beyond. The EVLA framework highlights the importance of considering capacity (biopsychosocial factors), capability (an individual’s understanding of cognitive and emotional, requirements), metacognitive factors (readiness- one’s awareness and effort to bring their capacity and capability to the task; action the composite of the behaviours that the individual produces to meet their task requirements in the moment) and continuity (the degree to which the appropriate behavioural response is maintained over time) [25]. While considering individual contributory factors, the framework includes consideration of system issues, the relationship between the individual, the system, and the interaction of the two and the likelihood the physician will repeat or modify his or her behaviour. Finally, the role of the system in contributing to the occurrence, maintenance or fuelling of behaviour, or conversely, its extinction, must be considered.

## 7. Conclusions

It is time we understood that medical professionalism and competence extends beyond knowledge and technical skills. For too long, as a profession, we have been guilty of excusing poor behaviour of physicians based on their being a “good clinician”, or otherwise serving the system by bringing in funds or supporting infrastructure. The aforementioned effects of dysfunctional or unprofessional behaviour on patient safety [6,29,30,31,32,33,39,40] makes it clear that a “good clinician with dysfunctional or unprofessional behaviour” is an oxymoron. Although identifying and addressing DPB is de rigueur in the US, and has been for over two decades, this is not the case elsewhere [8]. Moreover, much of the abundant literature pertaining to DPB is theoretical, with little empirical investigation of interventive strategies beyond the aforementioned remedial programmes teaching professionalism [44] and the skills training for peer intervention [38]. This warrants both future studies of this phenomenon in other countries, as well as more extensive empirical study of interventions, both at an individual level and at the systemic level. Future systemic research should investigate outcomes related to knowledge translation and awareness raising, as well as economic cost–benefit analyses and patient quality and safety outcomes.

In the absence of articulation of the concept of DPB outside the US, competence, professionalism, and fitness to practice are being defined and taught more holistically elsewhere [50,51], including in non-Western health settings [52]. Increasingly, dysfunctional or unprofessional behaviour per se can be grounds for notification to regulatory agencies, even for otherwise “good clinicians” [53,54]. For some, this is the only impetus for change, action, or treatment, as evidenced in the US. Given the poor self-awareness of clinicians with dysfunctional or unprofessional behaviour, this is not only a salve for beleaguered colleagues and the wider health system, but for the physicians themselves, whose behaviour may be under-pinned by treatable mental illness, trauma and/or adverse early life experiences [16]. Considering the catastrophic effects on the system and the physician themselves of inaction in response to DPB, we are strong advocates not only for real time intervention, but for prevention [55]. However, prevention and intervention are contingent upon the top-down systemic policy and managerial support we have observed in the US with systemic boundary and rule setting articulated by agencies such as the Joint Commission. We are not doing anyone any favours by ignoring, acting as bystanders, or otherwise turning a blind eye to dysfunctional or unprofessional behaviour; otherwise, we share culpability.

## Data Availability

Not applicable.

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
