# Peer review of "Pragmatic Systemic Solutions to the Wicked and Persistent Problem of the Unprofessional Disruptive Physician in the Health System"

_healthcare, 2023, doi:10.3390/healthcare11172455_

Round 1
Reviewer 1 Report
It would be significant if the medical system could control unethical and unprofessional actions/behaviors and disruptive behavior of medical staff. From this perspective, I think the meaning of this study is very high. However, this study is evaluated as a compilation of the author's opinions.
There are sentences in which the content of the introduction overlaps with the abstract, and the necessity and purpose of this study are not apparent.
Literature Review: It is necessary to review previous research on destructive behaviors or actions of physicians as medical staff. The authors should present their proposed perspectives on a systematic review of previous studies.
Results: Based on previous research, the obtained results and the specific points the authors intend to propose should be clearly stated.
Conclusion: The conclusion should address the aspects presented in this study and explain the academic and practical contributions of this research. Additionally, an explanation of the limitations of the study is necessary.
English Grammar: Some modifications for singular/plural nouns and sentence punctuation are required.
specific comments:
This study is evaluated as a research proposal to suggest how to deal with non-professional behavior among doctors within the medical system. However, there must be clear evidence of doctors' non-professional or disruptive behavior. Specific explanations about related research and cases are lacking, and it should also address from which perspective a particular action is considered non-professional or disruptive during patient care. In other words, a comprehensive review is necessary to adequately compare and examine the negative and positive aspects of medical staff behavior (e.g., non-professional or disruptive) during treatment.
Merely describing cultural change as a potential solution may not be sufficient. Healthcare is fundamentally based on reciprocity and equality and should be provided with respect for "human dignity." This is why the Nightingale Pledge exists. To emphasize cultural change, it is necessary to compare and examine cultural differences in each country and then provide a logical and evidence-based argument that medical services should be based on human dignity from an AAAA perspective, considering that cultural characteristics differ from country to country.
Additionally, more specific explanations are needed regarding the research objectives and a concrete description of the research outcomes for the conclusion. It should elucidate how these research results contribute academically and practically from different perspectives. Moreover, the research limitations and future research directions are insufficiently detailed.
Please note that some parts of the manuscript have been highlighted with potential grammatical errors.
“Cultural change is required to ensure systemic awareness and responsiveness. [6, 11, 34] We build on existing solutions [6], for recommendations to effect such change. First and foremost, as stated earlier, we must raise systemic awareness. There must be education of the organisation/hospital executive and administration and human resource department about DPB to enhance systemic responsiveness, facilitated by an understanding of the risks of inaction [6, 11, 34], with its converse, prevention and early intervention [38] at the core of all responses. Risks of inaction include compromise to patient satisfaction or organizational reputation, at best; or at worst, patient safety with serious adverse events including mortality, and resultant litigation [6, 29-33, 39-40]. Other systemic risks include effects on organizational morale, recruitment and retention, and care efficiency with compromised process flow and productivity [6, 29-33, 39-40], all of which carry additional
financial risks [41]. Besides these systemic risks, an oft-neglected risk conferred by failure to address DPB is the unaddressed and potential deterioration of mental health and wellbeing of the doctor with DPB, themselves often otherwise valuable assets of the system.”
It would be significant if the medical system could control unethical and unprofessional actions/behaviors and disruptive behavior of medical staff. From this perspective, I think the meaning of this study is very high. However, this study is evaluated as a compilation of the author's opinions.
There are sentences in which the content of the introduction overlaps with the abstract, and the necessity and purpose of this study are not apparent.
Literature Review: It is necessary to review previous research on destructive behaviors or actions of physicians as medical staff. The authors should present their proposed perspectives on a systematic review of previous studies.
Results: Based on previous research, the obtained results and the specific points the authors intend to propose should be clearly stated.
Conclusion: The conclusion should address the aspects presented in this study and explain the academic and practical contributions of this research. Additionally, an explanation of the limitations of the study is necessary.
English Grammar: Some modifications for singular/plural nouns and sentence punctuation are required.
Author Response
Editors and Reviewers, Healthcare
Re: Manuscript ID: healthcare-2481824
Type of manuscript: Commentary
Title: Pragmatic systemic solutions to the wicked and persistent problem of
the disruptive unprofessional physician in the health system
Authors: Carmelle Peisah *, Betsy Williams, Peter Hockey, Peter Lees, Danette
Wright, Alan Rosenstein
Thank you for giving us the opportunity to revise the above manuscript. We particularly wish to thank the Reviewers for their most helpful comments which we feel greatly enhance the manuscript. We have revised the manuscript accordingly:
Reviewer 1
1.1 Reviewer’s comment: It would be significant if the medical system could control unethical and unprofessional actions/behaviors and disruptive behavior of medical staff. From this perspective, I think the meaning of this study is very high. However, this study is evaluated as a compilation of the author's opinions.
Authors’ response Thank you very much for this comment. We note that this is a Commentary (as outlined in the Header Descriptor and in the Aims), not a study.
1.2 Reviewer’s comment. There are sentences in which the content of the introduction overlaps with the abstract, and the necessity and purpose of this study are not apparent.
Authors’ response We note that this is a Commentary (as outlined in the Header Descriptor and in the Aims) and not a study. Please see Page 2 Lines 64-86 where we extensively outline the aims, necessity and purpose of the Commentary.
1.3 Reviewer’s comment. Literature Review: It is necessary to review previous research on destructive behaviors or actions of physicians as medical staff. The authors should present their proposed perspectives on a systematic review of previous studies.
Authors’ response Please note this is a Commentary (as outlined in the Header Descriptor and in the Aims), not a Systematic Review. Notwithstanding this, the Commentary was a highly scholarly one (as described by Reviewer 4), supported by an extensive literature review, based on 52 references. Additionally, we have now discussed the paucity of empirical research in this field: Although identifying and addressing DPB is de rigueur in the US, and has been for over two decades, this is not the case elsewhere [8]. Moreover, much of the abundant literature pertaining to DPB is theoretical, with little empirical investigation of interventive strategies beyond the aforementioned remedial programmes teaching professionalism [44] and the skills training for peer intervention [38].
1.4 Reviewer’s comment. Results: Based on previous research, the obtained results and the specific points the authors intend to propose should be clearly stated.
Authors’ response: Please note this is a Commentary, (as outlined in the Header Descriptor and in the Aims) not a study. As such there are no results to be reported.
1.5 Reviewer’s comment: Conclusion: The conclusion should address the aspects presented in this study and explain the academic and practical contributions of this research. Additionally, an explanation of the limitations of the study is necessary.
Authors’ response: Please note this is a Commentary (as outlined in the Header Descriptor and in the Aims) not a research study. The practical contributions are outlined in Section 6 “Potential New Solutions” and in the Conclusion but these have now been elaborated upon.
1.6 Reviewer’s comment English Grammar: Some modifications for singular/plural nouns and sentence punctuation are required.
Authors’ response: Thank you for these suggestions, which we have adopted – see below.
1.7 Reviewer’s comment 1.7 specific comments:
This study is evaluated as a research proposal to suggest how to deal with non-professional behavior among doctors within the medical system. However, there must be clear evidence of doctors' non-professional or disruptive behavior. Specific explanations about related research and cases are lacking, and it should also address from which perspective a particular action is considered non-professional or disruptive during patient care. In other words, a comprehensive review is necessary to adequately compare and examine the negative and positive aspects of medical staff behavior (e.g., non-professional or disruptive) during treatment.
Authors’ response: Please note this is a Commentary (as outlined in the Header Descriptor and in the Aims) not a research proposal, nor a Systematic Review. Extensive explanations about related research have been provided. As we have stated in the article, the observation of this phenomenon and the development of the literature in this space, dating back over 150 years, means that the negative and positive aspects of physician behavior have been articulated extensively, thus rendering redundant comparisons of such.
1.8 Reviewer’s comment Merely describing cultural change as a potential solution may not be sufficient. Healthcare is fundamentally based on reciprocity and equality and should be provided with respect for "human dignity." This is why the Nightingale Pledge exists. To emphasize cultural change, it is necessary to compare and examine cultural differences in each country and then provide a logical and evidence-based argument that medical services should be based on human dignity from an AAAA perspective, considering that cultural characteristics differ from country to country.
Authors’ response: Thank you for these comments. We agree with these comments as they pertain to patient care and medical services. However, these are not the foci of this paper, nor is culture as it pertains to different countries, but rather medical culture particular to individual healthcare systems. We are unfamiliar with the “AAAA perspective”. We understand the Nightingale Pledge to be a statement of ethics and principles of the nursing profession in the United States, and while a vow to “abstain from whatever is deleterious and mischievous” is akin to what we are describing here, it is specific to, and perhaps has resonance for nursing staff from the US, with perhaps less salience to our paper which focuses on medical staff globally. We have now more clearly justified our focus on doctors in the introduction: Moreover, while we recognise that dysfunctional behaviour in health systems is not limited to physicians, over 80% of notifications for dysfunctional behaviour relate to physician behaviour, while 50% of notifications relate to nurses, with minimal study of other health disciplines.[8]
1.9 Reviewer’s comment Additionally, more specific explanations are needed regarding the research objectives and a concrete description of the research outcomes for the conclusion. It should elucidate how these research results contribute academically and practically from different perspectives. Moreover, the research limitations and future research directions are insufficiently detailed.
Authors’ response: Please note this is a Commentary (as outlined in the Header Descriptor and in the Aims) not a research proposal. Therefore, elucidation of results and research limitations are not applicable here. However, we have now outlined future research directions as suggested:
Although identifying and addressing DPB is de rigueur in the US, and has been for over two decades, this is not the case elsewhere [8]. Moreover, much of the abundant literature pertaining to DPB is theoretical, with little empirical investigation of interventive strategies beyond the aforementioned remedial programmes teaching professionalism [44] and the skills training for peer intervention [38]. This warrants both future studies of this phenomenon in other countries, as well as more extensive empirical study of interventions, both at an individual level and at the systemic level. Future systemic research should investigate outcomes related to knowledge translation and awareness raising, as well as economic cost- benefit analyses and patient quality and safety outcomes.
1.10 Reviewer’s comment Please note that some parts of the manuscript have been highlighted with potential grammatical errors.
“Cultural change is required to ensure systemic awareness and responsiveness. [6, 11, 34] We build on existing solutions [6], for recommendations to effect such change. First and foremost, as stated earlier, we must raise systemic awareness. There must be education of the organisation/hospital executive and administration and human resource department about DPB to enhance systemic responsiveness, facilitated by an understanding of the risks of inaction [6, 11, 34], with its converse, prevention and early intervention [38] at the core of all responses. Risks of inaction include compromise to patient satisfaction or organizational reputation, at best; or at worst, patient safety with serious adverse events including mortality, and resultant litigation [6, 29-33, 39-40]. Other systemic risks include effects on organizational morale, recruitment and retention, and care efficiency with compromised process flow and productivity [6, 29-33, 39-40], all of which carry additional financial risks [41]. Besides these systemic risks, an oft-neglected risk conferred by failure to address DPB is the unaddressed and potential deterioration of mental health and wellbeing of the doctor with DPB, themselves often otherwise valuable assets of the system.”
Authors’ response: Thank you for these suggestions and for the helpful highlighting. We have now addressed these as follows:-
“Cultural change is required to ensure systemic awareness and responsiveness. [6, 11, 34] We build on existing recommendations [6], to effect cultural change. First and foremost, as stated earlier, we must raise systemic awareness from the top of the healthcare organisation down. The organisation/hospital executive and administration, and human resource department must all be educated about DPB to enhance systemic responsiveness. Awareness raising of the risks of inaction towards DPB [6, 11, 34], with its converse, prevention and early intervention [38] may galvanise earlier action from the hospital administration. Risks of inaction include, at minimum, compromised patient satisfaction and organizational reputation with resultant litigation, and at maximum, risks to patient safety with serious adverse events, including patient mortality [6, 29-33, 39-40]. Other systemic risks include effects on organizational morale, recruitment and retention, and care efficiency with compromised process flow and productivity [6, 29-33, 39-40], all of which carry additional financial risks [41]. Besides these systemic risks, an oft-neglected risk conferred by failure to address DPB is the unaddressed and potential deterioration of mental health and wellbeing of the doctor with DPB, themselves often otherwise valuable assets of the system.
Reviewer 2 Report
Dear Authors,
I really appreciated your paper. Please find here below my comments to improve it:
1. I wonder if there are some evidences about the differences in different healthcare branches (internal medicine, surgery, intensive care, etc)
2. Is it possible to find some impact of disruptive professional behavior (DPB) in terms of economics of hospitals? In the paper the impact in terms of health are quoted but nothing in terms of cost increase.
3. Are the data from different countries being collected with the same methodology, can you say something more about the soundness of comparisons?
4. Can you say something more about the system factors you refer to? Do you mean incentivisation system, performance management system, performance evaluation systems, career development criteria or others
5. Can you add few lines about the individual factors? do you mean burnout effect, lack of updates.
6. In the conclusion, can you say something about policy implications and managerial implications? Which intervention can reduce the phenomenon?
Author Response
Editors and Reviewers, Healthcare
Re: Manuscript ID: healthcare-2481824
Type of manuscript: Commentary
Title: Pragmatic systemic solutions to the wicked and persistent problem of
the disruptive unprofessional physician in the health system
Authors: Carmelle Peisah *, Betsy Williams, Peter Hockey, Peter Lees, Danette
Wright, Alan Rosenstein
Thank you for giving us the opportunity to revise the above manuscript. We particularly wish to thank the Reviewers for their most helpful comments which we feel greatly enhance the manuscript. We have revised the manuscript accordingly:
Reviewer 2
Dear Authors,
2.1 Reviewer’s comment I really appreciated your paper. Please find here below my comments to improve it: I wonder if there are some evidences about the differences in different healthcare branches (internal medicine, surgery, intensive care, etc)
Authors’ response: Thankyou for these kind comments and for the helpful suggestion. We have now provided evidence for differences in the context of DPB (exactly as you predicted), with suggested reasons posed by the authors: Notably, the most common context for DPB are complex health environments with high care levels and associated emotional and physical burden, namely: intensive care, surgical and emergency departments [8].
2.2 Reviewer’s comment Is it possible to find some impact of disruptive professional behavior (DPB) in terms of economics of hospitals? In the paper the impact in terms of health are quoted but nothing in terms of cost increase.
Authors’ response: Thank you for this excellent suggestion. We had previously referred to: Other systemic risks include effects on organizational morale, recruitment and retention, and care efficiency with compromised process flow and productivity [6, 29-33, 39-40], all of which carry additional financial risks [41]. We have now elaborated on this and added another reference: One study of a 400-bed hospital estimated the combined costs of DPB attributable to medication and procedural errors, and staff turnover, were in excess of $1 million [42].
We thought that this was such an important issue that we also suggested future research in this regard: Future systemic research should investigate outcomes related to knowledge translation and awareness raising, as well as economic cost-benefit analyses and patient quality and safety outcomes.
2.3 Reviewer’s comment Are the data from different countries being collected with the same methodology, can you say something more about the soundness of comparisons?
Authors’ response: We have now outlined the comparative output regarding DPB across different countries. This not being a systematic review, we cannot comment on the soundness of comparisons or the methodology: Although identifying and addressing DPB is de rigueur in the US, and has been for over two decades, this is not the case elsewhere [8]. Moreover, much of the abundant literature pertaining to DPB is theoretical, with little empirical investigation of interventive strategies beyond the aforementioned remedial programmes teaching professionalism [44] and the skills training for peer intervention [38].
2.4 Reviewer’s comment 4. Can you say something more about the system factors you refer to? Do you mean incentivisation system, performance management system, performance evaluation systems, career development criteria or others
Authors’ response: We apologise for this confusion, recognising the myriad interpretation of the concept of system factors. We have now clarified the System Theory framework and elaborated on the Systemic Factors: The kind of systemic factors at play here include (i) the rules of the health system (e.g. “We have never heard of DPB” or “We tolerate DPB” or “We can’t touch her, she is too powerful to take on”); (ii) the history of the health system; and (iii) the functioning and structure of the health system, and relationships within, including alliances and conflicts. This broader perspective per se might help generate more solutions than reductionistic approaches to the causes of DPB, such as the simplistic linear, “individual x causes y” approach, which may limit options in response and quelch hope.
2.5 Reviewer’s comment Can you add few lines about the individual factors? do you mean burnout effect, lack of updates.
Authors’ response: The individual factors were extensively discussed in Section 2 but were obfuscated by the opening lines of this Section which digressed to Systemic factors. We apologise for the confusion created by this and have now removed the discussion of systemic factors to the introduction, using this as an opportunity also to explain further about Systemic factors (see above Response 2.4).
2.6 Reviewer’s comment In the conclusion, can you say something about policy implications and managerial implications? Which intervention can reduce the phenomenon?
Authors’ response: Thankyou for this suggestion – we have highlighted the paucity of evidence around interventions (see response to 3.3): Although identifying and addressing DPB is de rigueur in the US, and has been for over two decades, this is not the case elsewhere [8]. Moreover, much of the abundant literature pertaining to DPB is theoretical, with little empirical investigation of interventive strategies beyond the aforementioned remedial programmes teaching professionalism [44] and the skills training for peer intervention [38].
Additionally, we have augmented the Conclusion with reference to policy and managerial implications: However, prevention and intervention are contingent upon the top-down systemic policy and managerial support we have observed in the US with systemic boundary and rule setting articulated by agencies such as the Joint Commission.
Reviewer 3 Report
The paper presents a narrative review, addressing the DPB in health systems. Undoubtedly it is an Interesting paper. I have provided some additional feedback for the authors. The commentary is very well written, but additional aspects can be added to the paper.
Chapter 1 – Introduction
This specific chapter is well structured and stated clearly the topic under investigation. Nevertheless, we suggest that the DPB topic be deeply presented through a theoretical framework approach in order to allow non-practitioners to understand the concept.
In lines 54, 55, and 56 we think it is important to provide one or two other examples of professional categories where the DPB manifests.
Chapter 7 – Conclusion
It is suggested to include a more detailed conclusion regarding the actual interventive and preventive actions that can be set in place in order to mitigate the DPB. Taking into consideration that the method applied was a narrative review, it is recommended to leave some clues regarding future investigation paths, including suggestions of empirical studies that could address the conclusions.
Author Response
Reviewer 3
Editors and Reviewers, Healthcare
Re: Manuscript ID: healthcare-2481824
Type of manuscript: Commentary
Title: Pragmatic systemic solutions to the wicked and persistent problem of
the disruptive unprofessional physician in the health system
Authors: Carmelle Peisah *, Betsy Williams, Peter Hockey, Peter Lees, Danette
Wright, Alan Rosenstein
Thank you for giving us the opportunity to revise the above manuscript. We particularly wish to thank the Reviewers for their most helpful comments which we feel greatly enhance the manuscript. We have revised the manuscript accordingly:
3.1 Reviewer’s comment The paper presents a narrative review, addressing the DPB in health systems. Undoubtedly it is an Interesting paper. I have provided some additional feedback for the authors. The commentary is very well written, but additional aspects can be added to the paper.
Chapter 1 – Introduction
This specific chapter is well structured and stated clearly the topic under investigation. Nevertheless, we suggest that the DPB topic be deeply presented through a theoretical framework approach in order to allow non-practitioners to understand the concept.
Authors’ response: Thank you for these kind comments. We have now more deeply embedded our discussion of DPB within a systemic framework, with elaboration of this in the Introduction.
3.2 Reviewer’s comment In lines 54, 55, and 56 we think it is important to provide one or two other examples of professional categories where the DPB manifests.
Authors’ response: Thankyou for this suggestion, which we feel enhances the justification of the focus of the paper. We have now, as suggested, added over lines 54-56:
Moreover, while we recognise that dysfunctional behaviour in health systems is not limited to physicians, over 80% of notifications for dysfunctional behaviour relate to physician behaviour, while 50% of notifications relate to nurses, with minimal study of other health disciplines.[8]
3.3 Reviewer’s comment Chapter 7 – Conclusion
It is suggested to include a more detailed conclusion regarding the actual interventive and preventive actions that can be set in place in order to mitigate the DPB. Taking into consideration that the method applied was a narrative review, it is recommended to leave some clues regarding future investigation paths, including suggestions of empirical studies that could address the conclusions.
Authors’ response: Thank you for this excellent suggestion. We have extensively expanded the conclusion. We have both highlighted the paucity of empirical data pertaining to interventions and outlined future investigation paths:
Although identifying and addressing DPB is de rigueur in the US, and has been for over two decades, this is not the case elsewhere [8]. Moreover, much of the abundant literature pertaining to DPB is theoretical, with little empirical investigation of interventive strategies beyond the aforementioned remedial programmes teaching professionalism [44] and the skills training for peer intervention [38]. This warrants both future studies of this phenomenon in other countries, as well as more extensive empirical study of interventions, both at an individual level and at the systemic level. Future systemic research should investigate outcomes related to knowledge translation and awareness raising, as well as economic cost- benefit analyses and patient quality and safety outcomes.
We have also expanded on the conclusion with regards to managerial and policy implications (see Response 2.6 to Reviewer 2).
Reviewer 4 Report
This is an insightful and scholarly piece. It is well written and provocative. I look forward to citing this.
Minor: I note most of the literature referenced in this article comes from the UK or USA. Was this intentional? Is DPB being researched in other countries? Do they have similar or different ideas? It might be worth a comment.
minor: page 3 "Every physician needs" - this quote has no closing quotation marks.
minor: There are a couple of full stops that are on the wrong side of the in-text references.
Thanks for writing this. It's a delight to review it.
Author Response
Editors and Reviewers, Healthcare
Re: Manuscript ID: healthcare-2481824
Type of manuscript: Commentary
Title: Pragmatic systemic solutions to the wicked and persistent problem of
the disruptive unprofessional physician in the health system
Authors: Carmelle Peisah *, Betsy Williams, Peter Hockey, Peter Lees, Danette
Wright, Alan Rosenstein
Thank you for giving us the opportunity to revise the above manuscript. We particularly wish to thank the Reviewers for their most helpful comments which we feel greatly enhance the manuscript. We have revised the manuscript accordingly:
Reviewer 4
4.1 Reviewer’s comment This is an insightful and scholarly piece. It is well written and provocative. I look forward to citing this.
Authors’ response: Thank you. We very much appreciate, and are heartened by, these kind words.
4.2 Reviewer’s comment Minor: I note most of the literature referenced in this article comes from the UK or USA. Was this intentional? Is DPB being researched in other countries? Do they have similar or different ideas? It might be worth a comment.
Authors’ response: This is such an excellent and insightful comment. The answer to your questions are: Yes. It was intentional that most of the literature comes from the USA, and No, DPB is hardly researched in other countries. These issues prompted us to write the piece. We have now, thanks to your suggestion, used this as a prompt for future research and have added these comments: (line 387):
Although identifying and addressing DPB is de rigueur in the US, and has been for over two decades, this is not the case elsewhere [8]. Moreover, much of the abundant literature pertaining to DPB is theoretical, with little empirical investigation of interventive strategies beyond the aforementioned remedial programmes teaching professionalism [44] and the skills training for peer intervention [38]. This warrants both future studies of this phenomenon in other countries, as well as more extensive empirical study of interventions, both at an individual level and at the systemic level. Future systemic research should investigate outcomes related to knowledge translation and awareness raising, as well as economic cost- benefit analyses and patient quality and safety outcomes.
4.3 Reviewer’s comment minor: page 3 "Every physician needs" - this quote has no closing quotation marks.
Authors’ response: Thank you so much for picking this up. We have now rectified this, and have additionally italicized the quote given its length, to show more clearly that it is a quote.
4.4 Reviewer’s comment minor: There are a couple of full stops that are on the wrong side of the in-text references.
Authors’ response: Again, thank you for your assiduousness in identifying these errors. We have now corrected them: see Lines 101, 112, 135, 179, 199, 225, 256, 264,
4.5 Reviewer’s comment Thanks for writing this. It's a delight to review it.
Authors’ response: Thank you for these kind words.
Round 2
Reviewer 1 Report
Round 1: Reject
Round 2: Unfortunately, the comments from Round 1 have not been discussed sufficiently.
Round 1: Reject
Round 2: Unfortunately, the comments from Round 1 have not been discussed sufficiently.
Author Response
We consider that we discussed all comments from Round 1 extensively and systematically, in a nine-page, 24-item response, addressing every single comment from all four Reviewers. We very much appreciated the comments from the Reviewers and feel that the paper is greatly enhanced as a result. We noted that Reviewer 1 reviewed the paper on both occasions based on a misunderstanding of its nature and intent, misperceiving it as an empirical research study, not a Commentary.